# Wasserstein Distributionally Robust Kalman Filtering

**Soroosh Shafieezadeh-Abadeh**      **Viet Anh Nguyen**      **Daniel Kuhn**
École Polytechnique Fédérale de Lausanne, CH-1015 Lausanne, Switzerland
{soroosh.shafiee,viet-anh.nguyen,daniel.kuhn} @epfl.ch

**Peyman Mohajerin Esfahani**
Delft Center for Systems and Control, TU Delft, The Netherlands
P.MohajerinEsfahani@tudelft.nl

## Abstract

We study a distributionally robust mean square error estimation problem over a nonconvex Wasserstein ambiguity set containing only normal distributions. We show that the optimal estimator and the least favorable distribution form a Nash equilibrium. Despite the non-convex nature of the ambiguity set, we prove that the estimation problem is equivalent to a tractable convex program. We further devise a Frank-Wolfe algorithm for this convex program whose direction-searching subproblem can be solved in a quasi-closed form. Using these ingredients, we introduce a distributionally robust Kalman filter that hedges against model risk.

## 1  Introduction

The Kalman filter is the workhorse for the online tracking and estimation of a dynamical system's internal state based on indirect observations [1]. It has been applied with remarkable success in areas as diverse as automatic control, brain-computer interaction, macroeconomics, robotics, signal processing, weather forecasting and many more. The classical Kalman filter critically relies on the availability of an accurate state-space model and is therefore susceptible to model risk. This observation has led to several attempts to robustify the Kalman filter against modeling errors.

The $\mathcal{H}_\infty$-filter targets situations in which the statistics of the noise process is uncertain and where one aims to minimize the worst case instead of the variance of the estimation error [3, 26]. This filter bounds the $\mathcal{H}_\infty$-norm of the transfer function that maps the disturbances to the estimation errors. However, in transient operation, the desired $\mathcal{H}_\infty$-performance is lost, and the filter may diverge unless some (typically restrictive) positivity condition holds in each iteration. In set-valued estimation the disturbance vectors are modeled through bounded sets such as ellipsoids [4, 22]. In this framework, one attempts to construct the smallest ellipsoids around the state estimates that are consistent with the observations and the exogenous disturbance ellipsoids. However, the resulting robust filters ignore any distributional information and thus have a tendency to be over-conservative. A filter that is robust against more general forms of (set-based) model uncertainty was first studied in [19]. This filter iteratively minimizes the worst-case mean square error across all models in the vicinity of a nominal state space model. While performing well in the face of large uncertainties, this filter may be too conservative under small uncertainties. A generalized Kalman filter that addresses this shortcoming and strikes the balance between nominal and worst-case performance has been proposed in [25]. A risk-sensitive Kalman filter is obtained by minimizing the moment-generating function instead of the mean of the squared estimation error [24]. This risk-sensitive Kalman filter is equivalent to a distributionally robust filter proposed in [12], which minimizes the worst-case mean square error across all joint state-output distributions in a Kullback-Leibler (KL) ball around a nominal distribution. Extensions to more general $\tau$-divergence balls are investigated in [27].

In this paper we use ideas from distributionally robust optimization to design a Kalman-type filter that is immunized against model risk. Specifically, we assume that the joint distribution of the states and outputs is uncertain but known to reside in a given *ambiguity set* that contains all distributions in the proximity of the nominal distribution generated by a nominal state-space model. The ambiguity set thus reflects our level of (dis)trust in the nominal model. We then construct the most accurate filter under the least favorable distribution in this set. The hope is that hedging against the worst-case distribution has a regularizing effect and will lead to a filter that performs well under the unknown true distribution. Distributionally robust filters of this type have been studied in [7, 16] using uncertainty sets for the covariance matrix of the state vector and in [12, 27] using ambiguity sets defined via information divergences. Inspired by recent progress in data-driven distributionally robust optimization [14], we construct here the ambiguity set as a ball around the nominal distribution with respect to the type-2 Wasserstein distance. The Wasserstein distance has seen widespread application in machine learning [2, 6, 18], and an intimate relation between regularization and Wasserstein distributional robustness has been discovered in [21, 20, 23, 15]. Also, the Wasserstein distance is known to be more statistically robust than other information divergences [5].

We summarize our main contributions as follows:

- We introduce a distributionally robust mean square estimation problem over a nonconvex Wasserstein ambiguity set containing *normal* distributions only, and we demonstrate that the optimal estimator and the least favorable distribution form a Nash equilibrium.

- Leveraging modern reformulation techniques from [15], we prove that this problem is equivalent to a tractable convex program—despite the nonconvex nature of the underlying ambiguity set—and that the optimal estimator is an affine function of the observations.

- We devise an efficient Frank-Wolfe-type first-order method inspired by [10] to solve the resulting convex program. We show that the direction-finding subproblem can be solved in quasi-closed form, and we derive the algorithm's convergence rate.

- We introduce a Wasserstein distributionally robust Kalman filter that hedges against model risk. The filter can be computed efficiently by solving a sequence of robust estimation problems via the proposed Frank-Wolfe algorithm. Its performance is validated on standard test instances.

All proofs are relegated to Appendix A, and additional numerical results are reported in Appendix B.

**Notation:** For any $A \in \mathbb{R}^{d \times d}$ we use $\mathrm{Tr}\,[A]$ to denote the trace and $\|A\|$ to denote the spectral norm of $A$. By slight abuse of notation, the Euclidean norm of $v \in \mathbb{R}^d$ is also denoted by $\|v\|$. Moreover, $I_d$ stands for the identity matrix in $\mathbb{R}^{d \times d}$. For any $A, B \in \mathbb{R}^{d \times d}$, we use $\langle A, B \rangle = \mathrm{Tr}\,[A^\top B]$ to denote the trace inner product. The space of all symmetric matrices in $\mathbb{R}^{d \times d}$ is denoted by $\mathbb{S}^d$. We use $\mathbb{S}^d_+$ ($\mathbb{S}^d_{++}$) to represent the cone of symmetric positive semidefinite (positive definite) matrices in $\mathbb{S}^d$. For any $A, B \in \mathbb{S}^d$, the relation $A \succeq B$ ($A \succ B$) means that $A - B \in \mathbb{S}^d_+$ ($A - B \in \mathbb{S}^d_{++}$). Finally, the set of all normal distribution on $\mathbb{R}^d$ is denoted by $\mathcal{N}_d$.

## 2 Robust Estimation with Wasserstein Ambiguity Sets

Consider the problem of estimating a signal $x \in \mathbb{R}^n$ from a potentially noisy observation $y \in \mathbb{R}^m$. In practice, the joint distribution of $x$ and $y$ is never directly observable and thus fundamentally uncertain. This distributional uncertainty should be taken into account in the estimation procedure. In this paper, we model distributional uncertainty through an *ambiguity set* $\mathcal{P}$, that is, a family of distributions on $\mathbb{R}^d$, $d = n + m$, that are sufficiently likely to govern $x$ and $y$ in view of the available data or that are sufficiently close to a prescribed nominal distribution. We then seek a robust estimator that minimizes the worst-case mean square error across all distributions in the ambiguity set. In the following, we propose to use the Wasserstein distance in order to construct ambiguity sets.

**Definition 2.1** (Wasserstein distance). The type-2 Wasserstein distance between two distributions $\mathbb{Q}_1$ and $\mathbb{Q}_2$ on $\mathbb{R}^d$ is defined as

$$W_2(\mathbb{Q}_1, \mathbb{Q}_2) \triangleq \inf_{\pi \in \Pi(\mathbb{Q}_1, \mathbb{Q}_2)} \left\{ \left( \int_{\mathbb{R}^d \times \mathbb{R}^d} \|z_1 - z_2\|^2 \, \pi(\mathrm{d}\, z_1, \mathrm{d}\, z_2) \right)^{\frac{1}{2}} \right\}, \tag{1}$$

where $\Pi(\mathbb{Q}_1, \mathbb{Q}_2)$ is the set of all probability distributions on $\mathbb{R}^d \times \mathbb{R}^d$ with marginals $\mathbb{Q}_1$ and $\mathbb{Q}_2$.

**Proposition 2.2** ([9, Proposition 7]). *The type-2 Wasserstein distance between two normal distributions $\mathbb{Q}_1 = \mathcal{N}_d(\mu_1, \Sigma_1)$ and $\mathbb{Q}_2 = \mathcal{N}_d(\mu_2, \Sigma_2)$ with $\mu_1, \mu_2 \in \mathbb{R}^d$ and $\Sigma_1, \Sigma_2 \in \mathbb{S}_+^d$ equals*

$$W_2(\mathbb{Q}_1, \mathbb{Q}_2) = \sqrt{\|\mu_1 - \mu_2\|^2 + \mathrm{Tr}\left[\Sigma_1 + \Sigma_2 - 2\left(\Sigma_2^{\frac{1}{2}} \Sigma_1 \Sigma_2^{\frac{1}{2}}\right)^{\frac{1}{2}}\right]}.$$

Consider now a $d$-dimensional random vector $z = [x^\top, y^\top]^\top$ comprising the signal $x \in \mathbb{R}^n$ and the observation $y \in \mathbb{R}^m$, where $d = n + m$. For a given ambiguity set $\mathcal{P}$, the *distributionally robust minimum mean square error estimator* of $x$ given $y$ is a solution of the outer minimization problem in

$$\inf_{\psi \in \mathcal{L}} \sup_{\mathbb{Q} \in \mathcal{P}} \mathbb{E}^{\mathbb{Q}}\left[\|x - \psi(y)\|^2\right], \tag{2}$$

where $\mathcal{L}$ denotes the family of all measurable functions from $\mathbb{R}^m$ to $\mathbb{R}^n$. Problem (2) can be viewed as a zero-sum game between a statistician choosing the estimator $\psi$ and a fictitious adversary (or nature) choosing the distribution $\mathbb{Q}$. By construction, the minimax estimator performs best under the worst possible distribution $\mathbb{Q} \in \mathcal{P}$. From now on we assume that $\mathcal{P}$ is the Wasserstein ambiguity set

$$\mathcal{P} = \left\{\mathbb{Q} \in \mathcal{N}_d \; : \; W_2(\mathbb{Q}, \mathbb{P}) \le \rho\right\}, \tag{3}$$

which can be interpreted as a ball of radius $\rho \ge 0$ in the space of normal distributions. We will further assume that $\mathcal{P}$ is centered at a normal distribution $\mathbb{P} = \mathcal{N}_d(\mu, \Sigma)$ with covariance matrix $\Sigma \succ 0$.

Even though the Wasserstein ambiguity set $\mathcal{P}$ is nonconvex (as mixtures of normal distributions are generically not normal), we can prove a minimax theorem, which ensures that one may interchange the infimum and the supremum in (2) without affecting the problem's optimal value.

**Theorem 2.3** (Minimax theorem). *If $\mathcal{P}$ is a Wasserstein ambiguity set of the form (3), then*

$$\inf_{\psi \in \mathcal{L}} \sup_{\mathbb{Q} \in \mathcal{P}} \mathbb{E}^{\mathbb{Q}}\left[\|x - \psi(y)\|^2\right] = \sup_{\mathbb{Q} \in \mathcal{P}} \inf_{\psi \in \mathcal{L}} \mathbb{E}^{\mathbb{Q}}\left[\|x - \psi(y)\|^2\right]. \tag{4}$$

**Remark 2.4** (Connection to Bayesian estimation). The optimal solutions $\psi^\star$ and $\mathbb{Q}^\star$ of the two dual problems in (4) represent the minimax strategies of the statistician and nature, respectively. Theorem 2.3 implies that $(\psi^\star, \mathbb{Q}^\star)$ forms a saddle point (and thus a Nash equilibrium) of the underlying zero-sum game. Hence, the robust estimator $\psi^\star$ is also the optimal Bayesian estimator for the prior $\mathbb{Q}^\star$. For this reason, $\mathbb{Q}^\star$ is often referred to as the *least favorable prior* [11].

We now demonstrate that the minimax problem (2) is equivalent to a tractable convex program, whose solution allows us to recover both the optimal estimator $\psi^\star$ as well as the least favorable prior $\mathbb{Q}^\star$.

**Theorem 2.5** (Tractable reformulation). *The minimax problem (2) with the Wasserstein ambiguity set (3) centered at $\mathbb{P} = \mathcal{N}_d(\mu, \Sigma)$, $\underline{\sigma} \triangleq \lambda_{\min}(\Sigma) > 0$, is equivalent to the finite convex program*

$$
\begin{aligned}
\sup \quad & \mathrm{Tr}\left[S_{xx} - S_{xy} S_{yy}^{-1} S_{yx}\right] \\
\mathrm{s.\,t.} \quad & S = \begin{bmatrix} S_{xx} & S_{xy} \\ S_{yx} & S_{yy} \end{bmatrix} \in \mathbb{S}_+^d, \quad S_{xx} \in \mathbb{S}_+^n, \quad S_{yy} \in \mathbb{S}_+^m, \quad S_{xy} = S_{yx}^\top \in \mathbb{R}^{n \times m} \\
& \mathrm{Tr}\left[S + \Sigma - 2\left(\Sigma^{\frac{1}{2}} S \Sigma^{\frac{1}{2}}\right)^{\frac{1}{2}}\right] \le \rho^2, \quad S \succeq \underline{\sigma} I_d.
\end{aligned} \tag{5}
$$

*If $S^\star, S_{xx}^\star, S_{yy}^\star$ and $S_{xy}^\star$ is optimal in (5) and $\mu = [\mu_x^\top, \mu_y^\top]^\top$ for some $\mu_x \in \mathbb{R}^n$ and $\mu_y \in \mathbb{R}^m$, then the affine function $\psi^\star(y) = S_{xy}^\star (S_{yy}^\star)^{-1}(y - \mu_y) + \mu_x$ is the distributionally robust minimum mean square error estimator, and the normal distribution $\mathbb{Q}^\star = \mathcal{N}_d(\mu, S^\star)$ is the least favorable prior.*

Theorem 2.5 provides a tractable procedure for constructing a Nash equilibrium $(\psi^\star, S^\star)$ for the statistician's game against nature. Note that if $\rho = 0$, then $S^\star = \Sigma$ is the unique solution to (5). In this case the estimator $\psi^\star$ reduces to the Bayesian estimator corresponding to the nominal distribution $\mathbb{P} = \mathcal{N}_d(\mu, \Sigma)$. We emphasize that the choice of the Wasserstein radius $\rho$ may have a significant impact on the resulting estimator. In fact, this is a key distinguishing feature of the Wasserstein ambiguity set (3) with respect to other popular divergence-based ambiguity sets.

**Remark 2.6** (Divergence-based ambiguity sets). As a natural alternative, one could replace the Wasserstein distance in (3) with an information divergence. For example, ambiguity sets defined via $\tau$-divergences, which encapsulate the popular KL divergence as a special case, have been studied in [12, 27]. As shown in [12, Theorem 1] and [27, Theorem 2.1], the optimal estimator corresponding to any $\tau$-divergence ambiguity set always coincides with the Bayesian estimator for the nominal distribution $\mathbb{P} = \mathcal{N}_d(\mu, \Sigma)$ irrespective of $\rho$. Thus, in stark contrast to the setting considered here, the size of a $\tau$-divergence ambiguity set has no impact on the corresponding optimal estimator. Moreover, the least favorable prior $\mathbb{Q} = \mathcal{N}_d(\mu, S^\star)$ for a $\tau$-divergence ambiguity set always satisfies

$$S^\star = \begin{bmatrix} S_{xx}^\star & \Sigma_{xy} \\ \Sigma_{yx} & \Sigma_{yy} \end{bmatrix}. \tag{6}$$

Thus, in order to harm the statistician, nature only perturbs the second moments of the signal but sets all second moments of the observation as well as all cross moments to their nominal values.

**Example 2.7** (Impact of $\rho$ on the Nash equilibrium). We illustrate the dependence of the saddle point $(\psi^\star, \mathbb{Q}^\star)$ on the size $\rho$ of the ambiguity set in a 2-dimensional example. Suppose that the nominal distribution $\mathbb{P}$ of $[x, y] \in \mathbb{R}^2$ satisfies $\mu_x = \mu_y = 0$, $\Sigma_{xx} = \Sigma_{xy} = 1$ and $\Sigma_{yy} = 1.1$, implying that the noise $w \triangleq y - x$ and the signal $x$ are independent ($\mathbb{E}^{\mathbb{P}}[xw] = \Sigma_{xy} - \Sigma_{xx} = 0$). Figure 1 visualizes the canonical $90\%$ confidence ellipsoids of the the least favorable priors as well as the graphs of the optimal estimators for different sizes of the Wasserstein and KL ambiguity sets. As $\rho$ increases, the least favorable prior for the Wasserstein ambiguity set displays the following interesting properties: (i) the signal variance $S_{xx}^\star$ increases, (ii) the measurement variance $S_{yy}^\star$ decreases, (iii) the signal-measurement covariance $S_{xy}^\star$ decreases towards 0, and (iv) the noise variance $\mathbb{E}^{\mathbb{Q}^\star}[w^2] = S_{yy}^\star - 2S_{xy}^\star + S_{xx}^\star$ increases. Hence, (v) the signal-noise covariance $\mathbb{E}^{\mathbb{Q}^\star}[xw] = S_{xy}^\star - S_{xx}^\star$ decreases and is *negative* for all $\rho > 0$, and (vi) the optimal estimator $\psi^\star$ tends to the zero function. Note that the optimal estimator and the measurement variance remain constant in $\rho$ when working with a KL ambiguity set.

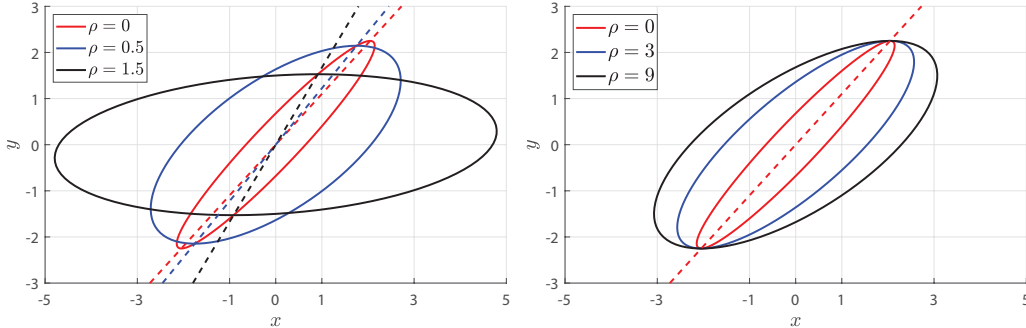

Figure 1: Least favorable priors (solid ellipsoids) and optimal estimators (dashed lines) for Wasserstein (left) and KL (right) ambiguity sets with different radii $\rho$. The Wasserstein estimators vary with $\rho$, while the KL estimators remain unaffected by $\rho$.

**Remark 2.8** (Ambiguity sets with non-normal distributions). Theorem 2.3 can be generalized to Wasserstein ambiguity set of the form $\mathcal{Q} = \{\mathbb{Q} \in \mathcal{M}(\mathbb{R}^d) : W_2(\mathbb{Q}, \mathbb{P}) \leq \rho\}$, where $\mathcal{M}(\mathbb{R}^d)$ denotes the set of all (possibly non-normal) probability distributions on $\mathbb{R}^d$ with finite second moments, and $\mathbb{P} = \mathcal{N}_d(\mu, \Sigma)$. In this case, the minimax result (4) remains valid provided that the set $\mathcal{L}$ of all measurable estimators is restricted to the set $\mathcal{A}$ of all affine estimators. Theorem 2.5 also remains valid under this alternative setting.

## 3 Efficient Frank-Wolfe Algorithm

The finite convex optimization problem (5) is numerically challenging as it constitutes a *nonlinear* semi-definite program (SDP). In principle, it would be possible to eliminate all nonlinearities by using Schur complements and to reformulate (5) as a *linear* SDP, which is formally tractable. However, it is folklore knowledge that general-purpose SDP solvers are yet to be developed that can reliably solve large-scale problem instances. We thus propose a tailored first-order method to solve the nonlinear SDP (5) directly, which exploits a covert structural property of the problem's objective function

$$f(S) \triangleq \mathrm{Tr}\left[S_{xx} - S_{xy}S_{yy}^{-1}S_{yx}\right].$$

**Definition 3.1** (Unit total elasticity[1]). We say that a function $\varphi : \mathbb{S}^d_+ \to \mathbb{R}_+$ has unit total elasticity if

$$\varphi(S) = \langle S, \nabla\varphi(S) \rangle \quad \forall S \in \mathbb{S}^d_+.$$

It is clear that every linear function has unit total elasticity. Maybe surprisingly, however, the objective function $f(S)$ of problem (5) also enjoys unit total elasticity because

$$\langle S, \nabla f(S) \rangle = \left\langle \begin{bmatrix} S_{xx} & S_{xy} \\ S_{yx} & S_{yy} \end{bmatrix}, \begin{bmatrix} I_n & -S_{xy}S_{yy}^{-1} \\ -S_{yy}^{-1}S_{yx} & S_{yy}^{-1}S_{yx}S_{xy}S_{yy}^{-1} \end{bmatrix} \right\rangle = f(S).$$

Moreover, as will be explained below, it turns out problem (5) can be solved highly efficiently if its objective function is replaced with a linear approximation. These observations motivate us to solve (5) with a Frank-Wolfe algorithm [8], which starts at $S^{(0)} = \Sigma$ and constructs iterates

$$S^{(k+1)} = \alpha_k F\big(S^{(k)}\big) + (1 - \alpha_k)S^{(k)} \quad \forall k \in \mathbb{N} \cup \{0\}, \tag{7a}$$

where $\alpha_k$ represents a judiciously chosen step-size, while the oracle mapping $F : \mathbb{S}_+ \to \mathbb{S}_+$ returns the unique solution of the direction-finding subproblem

$$F(S) \triangleq \begin{cases} \underset{L \succeq \underline{\sigma} I_d}{\arg\max} & \langle L, \nabla f(S) \rangle \\ \text{s.t.} & \mathrm{Tr}\left[ L + \Sigma - 2 \left( \Sigma^{\frac{1}{2}} L \Sigma^{\frac{1}{2}} \right)^{\frac{1}{2}} \right] \leq \rho^2 . \end{cases} \tag{7b}$$

In each iteration, the Frank-Wolfe algorithm thus maximizes a linearized objective function over the original feasible set. In contrast to other commonly used first-order methods, the Frank-Wolfe algorithm thus obviates the need for a potentially expensive projection step to recover feasibility. It is easy to convince oneself that any solution of the nonlinear SDP (5) is indeed a fixed point of the operator $F$. To make the Frank-Wolfe algorithm (7) work in practice, however, one needs

   (i) an efficient routine for solving the direction-finding subproblem (7b);

   (ii) a step-size rule that offers rigorous guarantees on the algorithm's convergence rate.

In the following, we propose an efficient bisection algorithm to address (i). As for (ii), we show that the convergence analysis portrayed in [10] applies to the problem at hand. The procedure for solving (7b) is outlined in Algorithm 1, which involves an auxiliary function $h : \mathbb{R}_+ \to \mathbb{R}$ defined via

$$h(\gamma) \triangleq \rho^2 - \left\langle \Sigma, \big( I_d - \gamma(\gamma I_d - \nabla f(S))^{-1} \big)^2 \right\rangle. \tag{8}$$

**Theorem 3.2** (Direction-finding subproblem). For any fixed inputs $\rho, \varepsilon \in \mathbb{R}_{++}$, $\Sigma \in \mathbb{S}^d_{++}$ and $S \in \mathbb{S}^d_+$, Algorithm 1 outputs a feasible and $\varepsilon$-suboptimal solution to (7b).

We emphasize that the most expensive operation in Algorithm 1 is the matrix inversion $(\gamma I_d - D)^{-1}$, which needs to be evaluated repeatedly for different values of $\gamma$. These computations can be accelerated by diagonalizing $D$ only once at the beginning. The repeat loop in Algorithm 1 carries out the actual bisection algorithm, and a suitable initial bisection interval is determined by a pair of a priori bounds $LB$ and $UB$, which are available in closed form (see Appendix A).

The overall structure of the proposed Frank-Wolfe method is summarized in Algorithm 2. We borrow the step-size rule suggested in [10] to establish rigorous convergence guarantees. This is accomplished by showing that the objective function $f$ has a bounded *curvature constant*. Our convergence result is formalized in the next theorem.

**Theorem 3.3** (Convergence analysis). If $\Sigma \succ 0$, $\rho > 0$, $\delta > 0$ and $\alpha_k = 2/(2 + k)$ for any $k \in \mathbb{N}$, then the $k$-th iterate $S^{(k)}$ computed by Algorithm 2 is feasible in (5) and satisfies

$$f(S^\star) - f(S^{(k)}) \leq \frac{4\bar{\sigma}^4}{\underline{\sigma}^3(k+2)}(1 + \delta),$$

where $S^\star$ is an optimal solution of (5), $\underline{\sigma}$ is the smallest eigenvalue of $\Sigma$, and $\bar{\sigma} \triangleq (\rho + \sqrt{\mathrm{Tr}\,[\Sigma]})^2$.

| **Algorithm 1** Bisection algorithm to solve (7b) | **Algorithm 2** Frank-Wolfe algorithm to solve (5) |
|---|---|
| **Input:** Covariance matrix $\Sigma \succ 0$ | **Input:** Covariance matrix $\Sigma \succ 0$ |
| $\quad$ Gradient matrix $D \triangleq \nabla f(S) \succeq 0$ | $\quad$ Wasserstein radius $\rho > 0$ |
| $\quad$ Wasserstein radius $\rho > 0$ | $\quad$ Tolerance $\delta > 0$ |
| $\quad$ Tolerance $\varepsilon > 0$ | Set $\underline{\sigma} \leftarrow \lambda_{\min}(\Sigma), \bar{\sigma} \leftarrow (\rho + \sqrt{\mathrm{Tr}\,[\Sigma]})^2$ |
| Denote the largest eigenvalue of $D$ by $\lambda_1$ | Set $\overline{C} \leftarrow 2\bar{\sigma}^4/\underline{\sigma}^3$ |
| Let $v_1$ be an eigenvector of $\lambda_1$ | Set $S^{(0)} \leftarrow \Sigma, k \leftarrow 0$ |
| Set $LB \leftarrow \lambda_1(1 + \sqrt{v_1^\top \Sigma v_1}/\rho)$ | **while** Stopping criterion is not met **do** |
| Set $UB \leftarrow \lambda_1(1 + \sqrt{\mathrm{Tr}\,[\Sigma]}/\rho)$ | $\quad$ Set $\alpha_k \leftarrow \frac{2}{k+2}$ |
| **repeat** | $\quad$ Set $G \leftarrow S_{xy}^{(k)}(S_{yy}^{(k)})^{-1}$ |
| $\quad$ Set $\gamma \leftarrow (UB + LB)/2$ | $\quad$ Compute gradient $D \leftarrow \nabla f(S^{(k)})$ by |
| $\quad$ Set $L \leftarrow \gamma^2(\gamma I_d - D)^{-1}\Sigma(\gamma I_d - D)^{-1}$ | $\quad\quad D \leftarrow [I_n, \ -G]^\top[I_n, \ -G]$ |
| $\quad$ **if** $h(\gamma) < 0$ **then** | $\quad$ Set $\varepsilon \leftarrow \alpha_k \delta \overline{C}$ |
| $\quad\quad$ Set $LB \leftarrow \gamma$ | $\quad$ Solve the subproblem (7b) by Algorithm 1 |
| $\quad$ **else** | $\quad\quad L \leftarrow \mathrm{Bisection}(\Sigma, D, \rho, \varepsilon)$ |
| $\quad\quad$ Set $UB \leftarrow \gamma$ | $\quad$ Set $S^{(k+1)} \leftarrow S^{(k)} + \alpha_k(L - S^{(k)})$ |
| $\quad$ **end if** | $\quad$ Set $k \leftarrow k + 1$ |
| $\quad$ Set $\Delta \leftarrow \gamma(\rho^2 - \mathrm{Tr}\,[\Sigma]) - \langle L, D \rangle$ | **end while** |
| $\quad\quad + \gamma^2 \langle (\gamma I_d - D)^{-1}, \Sigma \rangle$ | |
| **until** $h(\gamma) > 0$ and $\Delta < \varepsilon$ | |
| **Output:** $L$ | **Output:** $S^{(k)}$ |

## 4 The Wasserstein Distributionally Robust Kalman Filter

Consider a discrete-time dynamical system whose (unobservable) state $x_t \in \mathbb{R}^n$ and (observable) output $y_t \in \mathbb{R}^m$ evolve randomly over time. At any time $t \in \mathbb{N}$, we aim to estimate the current state $x_t$ based on the output history $Y_t \triangleq (y_1, \ldots, y_t)$. We assume that the joint state-output process $z_t = [x_t^\top, \ y_t^\top]^\top$, $t \in \mathbb{N}$, is governed by an unknown Gaussian distribution $\mathbb{Q}$ in the neighborhood of a known nominal distribution $\mathbb{P}^\star$. The distribution $\mathbb{P}^\star$ is determined through the linear state-space model

$$\left.\begin{array}{l} x_t = A_t x_{t-1} + B_t v_t \\ y_t = C_t x_t + D_t v_t \end{array}\right\} \quad \forall t \in \mathbb{N}, \tag{9}$$

where $A_t$, $B_t$, $C_t$, and $D_t$ are given matrices of appropriate dimensions, while $v_t \in \mathbb{R}^d$, $t \in \mathbb{N}$, denotes a Gaussian white noise process independent of $x_0 \sim \mathcal{N}_n(\hat{x}_0, V_0)$. Thus, $v_t \sim \mathcal{N}_d(0, I_d)$ for all $t$, while $v_t$ and $v_{t'}$ are independent for all $t \neq t'$. Note that we may restrict the dimension of $v_t$ to the dimension $d = n + m$ of $z_t$ without loss of generality. Otherwise, all linearly dependent columns of $[B_t^\top, \ D_t^\top]^\top$ and the corresponding components of $v_t$ can be eliminated systematically.

By the law of total probability and the Markovian nature of the state-space model (9), the nominal distribution $\mathbb{P}^\star$ is uniquely determined by the marginal distribution $\mathbb{P}_{x_0}^\star = \mathcal{N}_n(\hat{x}_0, V_0)$ of the initial state $x_0$ and the conditional distributions

$$\mathbb{P}_{z_t|x_{t-1}}^\star = \mathcal{N}_d\left(\begin{bmatrix} A_t \\ C_t A_t \end{bmatrix} x_{t-1}, \begin{bmatrix} B_t \\ C_t B_t + D_t \end{bmatrix}\begin{bmatrix} B_t \\ C_t B_t + D_t \end{bmatrix}^\top\right)$$

of $z_t$ given $x_{t-1}$ for all $t \in \mathbb{N}$.

Unlike $\mathbb{P}^\star$, the true distribution $\mathbb{Q}$ governing $z_t$, $t \in \mathbb{N}$, is unknown, and thus the estimation problem at hand is not well-defined. We will therefore estimate the conditional mean $\hat{x}_t$ and covariance matrix $V_t$ of $x_t$ given $Y_t$ under some worst-case distribution $\mathbb{Q}^\star$ to be constructed recursively. First, we assume that the marginal distribution $\mathbb{Q}_{x_0}^\star$ of $x_0$ under $\mathbb{Q}^\star$ equals $\mathbb{P}_{x_0}$, that is, $\mathbb{Q}_{x_0}^\star = \mathcal{N}_n(\hat{x}_0, V_0)$. Next, fix any $t \in \mathbb{N}$ and assume that the conditional distribution $\mathbb{Q}_{x_{t-1}|Y_{t-1}}^\star$ of $x_{t-1}$ given $Y_{t-1}$ under $\mathbb{Q}^\star$ has already been computed as $\mathbb{Q}_{x_{t-1}|Y_{t-1}}^\star = \mathcal{N}_n(\hat{x}_{t-1}, V_{t-1})$. The construction of $\mathbb{Q}_{x_t|Y_t}^\star$ is then split into a prediction step and an update step. The prediction step combines the previous state estimate $\mathbb{Q}_{x_{t-1}|Y_{t-1}}^\star$ with the nominal transition kernel $\mathbb{P}_{z_t|x_{t-1}}^\star$ to generate a pseudo-nominal

---

**Algorithm 3** Robust Kalman filter at time $t$

---

**Input:** Covariance matrix $V_{t-1} \succeq 0$
       State estimate $\hat{x}_{t-1}$
       Wasserstein radius $\rho_t > 0$
       Tolerance $\delta > 0$
  **Prediction:**
       Form the pseudo-nominal distribution
       $\mathbb{P}_{z_t|Y_{t-1}} = \mathcal{N}_d(\mu_t, \Sigma_t)$ using (10)
  **Observation:**
       Observe the output $y_t$
  **Update:**
       Use Algorithm 2 to solve (11)
          $S_t^\star \leftarrow$ Frank-Wolfe$(\Sigma_t, \mu_t, \rho_t, \delta)$
**Output:** $V_t = S_{t,xx} - S_{t,xy}(S_{t,yy})^{-1}S_{t,yx}$
         $\hat{x}_t = S_{t,xy}^\star(S_{t,yy}^\star)^{-1}(y_t - \mu_{t,y}) + \mu_{t,x}$

---

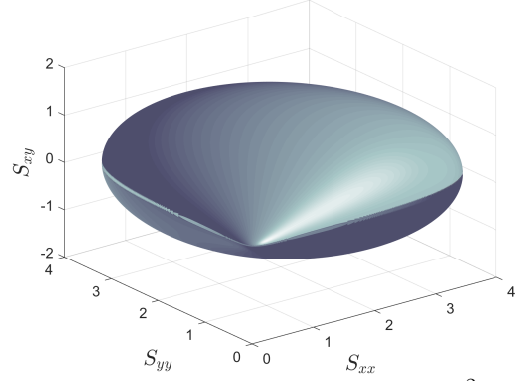

Figure 2: Wasserstein ball in the space $\mathbb{S}_+^2$ of covariance matrices centered at $I_2$ with radius 1.

distribution $\mathbb{P}_{z_t|Y_{t-1}}$ of $z_t$ conditioned on $Y_{t-1}$, which is defined through

$$\mathbb{P}_{z_t|Y_{t-1}}(B|Y_{t-1}) = \int_{\mathbb{R}^n} \mathbb{P}_{z_t|x_{t-1}}^\star(B|x_{t-1})\mathbb{Q}_{x_{t-1}|Y_{t-1}}^\star(\mathrm{d}x_{t-1}|Y_{t-1})$$

for every Borel set $B \subseteq \mathbb{R}^d$ and observation history $Y_{t-1} \in \mathbb{R}^{m \times (t-1)}$. The well-known formula for the convolution of two multivariate Gaussians reveals that $\mathbb{P}_{z_t|Y_{t-1}} = \mathcal{N}_d(\mu_t, \Sigma_t)$, where

$$\mu_t = \begin{bmatrix} A_t \\ C_t A_t \end{bmatrix} \hat{x}_{t-1} \quad \text{and} \quad \Sigma_t = \begin{bmatrix} A_t \\ C_t A_t \end{bmatrix} V_{t-1} \begin{bmatrix} A_t \\ C_t A_t \end{bmatrix}^\top + \begin{bmatrix} B_t \\ C_t B_t + D_t \end{bmatrix} \begin{bmatrix} B_t \\ C_t B_t + D_t \end{bmatrix}^\top . \quad (10)$$

Note that the construction of $\mathbb{P}_{z_t|Y_{t-1}}$ resembles the prediction step of the classical Kalman filter but uses the least favorable distribution $\mathbb{Q}_{x_{t-1}|Y_{t-1}}^\star$ instead of the nominal distribution $\mathbb{P}_{x_{t-1}|Y_{t-1}}^\star$.

In the update step, the pseudo-nominal a priori estimate $\mathbb{P}_{z_t|Y_{t-1}}$ is updated by the measurement $y_t$ and robustified against model uncertainty to yield a refined a posteriori estimate $\mathbb{Q}_{x_t|Y_t}^\star$. This a posteriori estimate is found by solving the minimax problem

$$\inf_{\psi_t \in \mathcal{L}} \sup_{\mathbb{Q} \in \mathcal{P}_{z_t|Y_{t-1}}} \mathbb{E}^{\mathbb{Q}}\left[\|x_t - \psi_t(y_t)\|^2\right] \quad (11)$$

equipped with the Wasserstein ambiguity set

$$\mathcal{P}_{z_t|Y_{t-1}} = \left\{\mathbb{Q} \in \mathcal{N}_d : W_2(\mathbb{Q}, \mathbb{P}_{z_t|Y_{t-1}}) \leq \rho_t\right\}.$$

Note that the Wasserstein radius $\rho_t$ quantifies our distrust in the pseudo-nominal a priori estimate and can therefore be interpreted as a measure of model uncertainty. Practically, we reformulate (11) as an equivalent finite convex program of the form (5), which is amenable to efficient computational solution via the Frank-Wolfe algorithm detailed in Section 3. By Theorem 2.5, the optimal solution $S_t^\star$ of problem (5) yields the least favorable conditional distribution $\mathbb{Q}_{z_t|Y_{t-1}}^\star = \mathcal{N}_d(\mu_t, S_t^\star)$ of $z_t$ given $Y_{t-1}$. By using the well-known formulas for conditional normal distributions (see, *e.g.*, [17, page 522]), we then obtain the least favorable conditional distribution $\mathbb{Q}_{x_t|Y_t}^\star = \mathcal{N}_n(\hat{x}_t, V_t)$ of $x_t$ given $Y_t$, where

$$\hat{x}_t = S_{t,xy}^\star(S_{t,yy}^\star)^{-1}(y_t - \mu_{t,y}) + \mu_{t,x} \quad \text{and} \quad V_t = S_{t,xx}^\star - S_{t,xy}^\star(S_{t,yy}^\star)^{-1}S_{t,yx}^\star.$$

The distributionally robust Kalman filtering approach is summarized in Algorithm 3. Note that the robust update step outlined above reduces to the usual update step of the classical Kalman filter for $\rho \downarrow 0$.

## 5 Numerical Results

We showcase the performance of the proposed Frank-Wolfe algorithm and the distributionally robust Kalman filter in a suite of synthetic experiments. All optimization problems are implemented in MATLAB and run on an Intel XEON CPU with 3.40GHz clock speed and 16GB of RAM, and the corresponding codes are made publicly available at https://github.com/sorooshafiee/WKF.

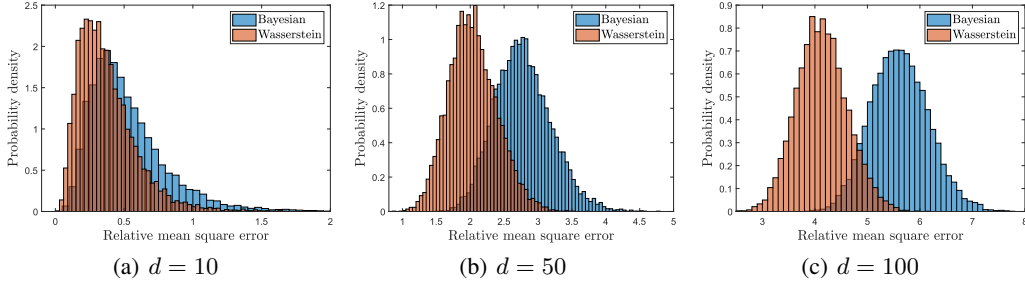

|  (a) $d = 10$ | (b) $d = 50$ | (c) $d = 100$ |

Figure 3: Distribution of the difference between the errors of the robust MMSE (Bayesian MMSE) and the ideal MMSE$^\star$ estimator.

## 5.1 Distributionally Robust Minimum Mean Square Error Estimation

We first assess the distributionally robust minimum mean square error (robust MMSE) estimator, which is obtained by solving (2), against the classical Bayesian MMSE estimator, which can be viewed as the solution of problem (2) over a singleton ambiguity set that contains only the nominal distribution. Recall from Remark 2.6 that the optimal estimator corresponding to a KL or $\tau$-divergence ambiguity set of the type studied in [12, 27] coincides with the Bayesian MMSE estimator irrespective of $\rho$. Thus, we may restrict attention to Wasserstein ambiguity sets. In order to develop a geometric intuition, Figure 2 visualizes the set of all bivariate normal distributions with zero mean that have a Wasserstein distance of at most 1 from the standard normal distribution—projected to the space of covariance matrices.

In the first experiment we aim to predict a signal $x \in \mathbb{R}^{4d/5}$ from an observation $y \in \mathbb{R}^{d/5}$, where the random vector $z = [x^\top, \; y^\top]^\top$ follows a $d$-variate Gaussian distribution with $d \in \{10, 50, 100\}$. The experiment comprises $10^4$ simulation runs. In each run we randomly generate two covariance matrices $\Sigma^\star$ and $\Sigma$ as follows. First, we draw two matrices $A^\star$ and $A$ from the standard normal distribution on $\mathbb{R}^{d \times d}$, and we denote by $R^\star$ and $R$ the orthogonal matrices whose columns correspond to the orthonormal eigenvectors of $A^\star + (A^\star)^\top$ and $A + A^\top$, respectively. Then, we define $\Delta^\star = R^\star \Lambda^\star (R^\star)^\top$ and $\Sigma = R \Lambda R^\top$, where $\Lambda^\star$ and $\Lambda$ are diagonal matrices whose main diagonals are sampled uniformly from $[0, 1]^d$ and $[0.1, 10]^d$, respectively. Finally, we set $\Sigma^\star = (\Sigma^{\frac{1}{2}} + (\Delta^\star)^{\frac{1}{2}})^2$ and define the normal distributions $\mathbb{P}^\star = \mathcal{N}_d(0, \Sigma^\star)$ and $\mathbb{P} = \mathcal{N}_d(0, \Sigma)$. By construction, we have

$$W_2(\mathbb{P}^\star, \mathbb{P}) \le \|(\Sigma^\star)^{\frac{1}{2}} - \Sigma^{\frac{1}{2}}\|_F \le \sqrt{d},$$

where $\|\cdot\|_F$ stands for the Frobenius norm, and the first inequality follows from [13, Proposition 3]. We assume that $\mathbb{P}^\star$ is the true distribution and $\mathbb{P}$ our nominal prior. The robust MMSE estimator is obtained by solving (5) for $\rho = \sqrt{d}$ via the Frank-Wolfe algorithm from Section 3, while the Bayesian MMSE estimator under $\mathbb{P}$ is calculated analytically. In order to provide a meaningful comparison between these two approaches, we also compute the Bayesian MMSE estimator under the true distribution $\mathbb{P}^\star$ (denoted by MMSE$^\star$), which is indeed the best possible estimator. Figure 3 visualizes the distribution of the difference between the mean square errors under $\mathbb{P}^\star$ of the robust MMSE (Bayesian MMSE) and MMSE$^\star$ estimators. We observe that the robust MMSE estimator produces better results consistently across all experiments, and the effect is more pronounced for larger dimensions $d$. Figures 4(a) and 4(b) report the execution time and the iteration complexity of the Frank-Wolfe algorithm for $d \in \{10, \ldots, 100\}$ when the algorithm is stopped as soon as the relative duality gap $\langle F(S^k) - S^k, \nabla f(S^k) \rangle / f(S^k)$ drops below $0.01\%$. Note that the execution time grows polynomially due to the matrix inversion in the bisection algorithm. Figure 4(c) shows the relative duality gap of the current solution as a function of the iteration count.

## 5.2 Wasserstein Distributionally Robust Kalman Filtering

We assess the performance of the proposed Wasserstein distributionally robust Kalman filter against that of the classical Kalman filter and the Kalman filter with the KL ambiguity set from [12]. To this end, we borrow the standard test instance from [19, 25, 12] with $n = 2$ and $m = 1$. The system matrices satisfy

$$A_t = \begin{bmatrix} 0.9802 & 0.0196 + 0.099\Delta_t \\ 0 & 0.9802 \end{bmatrix}, \; B_t B_t^\top = \begin{bmatrix} 1.9608 & 0.0195 \\ 0.0195 & 1.9605 \end{bmatrix}, \; C_t = [1, \; -1], \; D_t D_t^\top = 1,$$

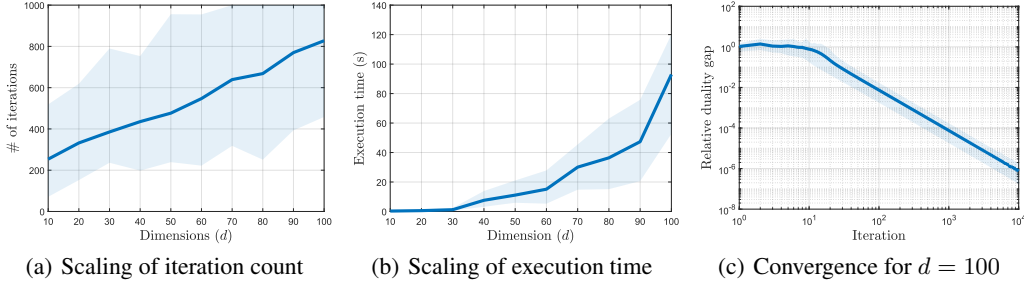

(a) Scaling of iteration count      (b) Scaling of execution time      (c) Convergence for $d = 100$

Figure 4: Convergence behavior of the Frank-Wolfe algorithm (shown are the average (solid line) and the range (shaded area) of the respective performance measures across 100 simulation runs)

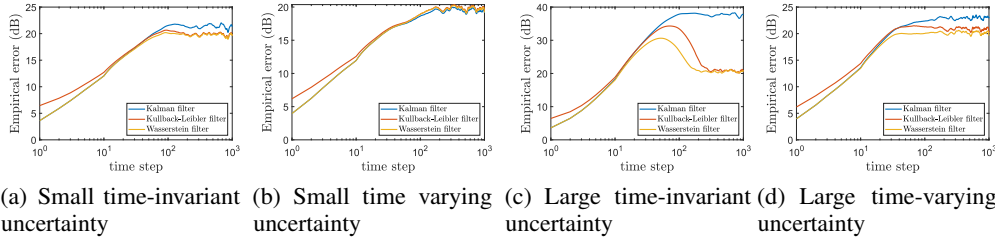

(a) Small time-invariant uncertainty      (b) Small time varying uncertainty      (c) Large time-invariant uncertainty      (d) Large time-varying uncertainty

Figure 5: Empirical means square estimation error of different filters

and $B_t D_t^\top = 0$, where $\Delta_t$ represents a scalar uncertainty, and the initial state satisfies $x_0 \sim \mathcal{N}_2(0, I_2)$. In all numerical experiments we simulate the different filters over 1000 periods starting from $\hat{x}_0 = 0$ and $V_0 = I_2$. Figure 5 shows the empirical mean square error $\frac{1}{500} \sum_{j=1}^{500} \|x_t^j - \hat{x}_t^j\|^2$ across 500 independent simulation runs, where $\hat{x}_t^j$ denotes the state estimate at time $t$ in the $j^{\text{th}}$ run. We distinguish four different scenarios: time-invariant uncertainty ($\Delta_t^j = \Delta^j$ sampled uniformly from $[-\bar{\Delta}, \bar{\Delta}]$ for each $j$) versus time-varying uncertainty ($\Delta_t^j$ sampled uniformly from $[-\bar{\Delta}, \bar{\Delta}]$ for each $t$ and $j$), and small uncertainty ($\bar{\Delta} = 1$) versus large uncertainty ($\bar{\Delta} = 10$). All results are reported in decibel units ($10 \log_{10}(\cdot)$). As for the filter design, the Wasserstein and KL radii are selected from the search grids $\{a \cdot 10^{-1} : a \in \{1, 1.1, \cdots, 2\}\}$ and $\{a \cdot 10^{-4} : a \in \{1, 1.1, \cdots, 2\}\}$, respectively. Figure 5 reports the results with minimum steady state error across all candidate radii.

Under small time-invariant uncertainty (Figure 5(a)), the Wasserstein and KL distributionally robust filters display a similar steady-state performance but outperform the classical Kalman filter. Note that the KL distributionally robust filter starts from a different initial point as we use the delayed implementation from [12]. Under small time-varying uncertainty (Figure 5(b)), both distributionally robust filters display a similar performance as the classical Kalman filter. Figures 5(c) and (d) corresponding to the case of large uncertainty are similar to Figures 5(a) and (b), respectively. However, the Wasserstein distributionally robust filter now significantly outperforms the classical Kalman filter and, to a lesser extent, the KL distributionally robust filter. Moreover, the Wasserstein distributionally robust filter exhibits the best transient behavior.

**Acknowledgments**   We gratefully acknowledge financial support from the Swiss National Science Foundation under grant BSCGI0_157733.

## Footnotes

[1]Our terminology is inspired by the definition of the elasticity of a univariate function $\varphi(s)$ as $\frac{\mathrm{d}\varphi(s)}{\mathrm{d}s} \frac{s}{\varphi(s)}$.

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
