[Supplementary Material · supplementary.pdf]

# Appendix A    Proofs

## A.1    Proof of Theorem 2.3

The proof of Theorem 2.3 requires the following preparatory lemma, which we borrow from [6].

**Lemma A.1** ([6, Proposition 2.8]).  *For any $\gamma \in \mathbb{R}_+$, $D \in \mathbb{S}_+^d \backslash \{0\}$ and $\Sigma \in \mathbb{S}_{++}^d$, we have*

$$\sup_{S \succeq 0} \left\langle D, S \right\rangle - \gamma \operatorname{Tr}\left[ S - 2\left( \Sigma^{\frac{1}{2}} S \Sigma^{\frac{1}{2}} \right)^{\frac{1}{2}} \right] = \begin{cases} \gamma^2 \left\langle (\gamma I_d - D)^{-1}, \Sigma \right\rangle & \text{if } \gamma I_d \succ D, \\ +\infty & \text{otherwise.} \end{cases}$$

*Moreover, if $\gamma I_d \succ D$, the unique optimal solution of the above maximization problem is given by*

$$S^\star = \gamma^2 (\gamma I_d - D)^{-1} \Sigma (\gamma I_d - D)^{-1}.$$

*Proof of Theorem 2.3.* The optimal value of the minimax problem (2) satisfies

$$\inf_{\psi \in \mathcal{L}} \sup_{\mathbb{Q} \in \mathcal{P}} \mathbb{E}^{\mathbb{Q}} \left[ \|x - \psi(y)\|^2 \right] \geq \sup_{\mathbb{Q} \in \mathcal{P}} \inf_{\psi \in \mathcal{L}} \mathbb{E}^{\mathbb{Q}} \left[ \|x - \psi(y)\|^2 \right] \tag{A.1a}$$

$$= \sup_{\mathbb{Q} \in \mathcal{P}} \inf_{G,g} \mathbb{E}^{\mathbb{Q}} \left[ \|x - Gy - g\|^2 \right], \tag{A.1b}$$

where (A.1a) follows from the max-min inequality, and (A.1b) holds because the inner minimization problem over $\psi$ is solved by the conditional expectation function $\psi^\star(y) = \mathbb{E}^{\mathbb{Q}}[x|y]$, which is affine in $y$ for every fixed Gaussian distribution $\mathbb{Q} \in \mathcal{P}$, see, *e.g.*, [7, page 522]. Without loss of generality, one can thus restrict the set of measurable functions $\mathcal{L}$ to the set of affine functions parametrized by a sensitivity matrix $G \in \mathbb{R}^{n \times m}$ and an intercept vector $g \in \mathbb{R}^n$. Recalling the definition of the Wasserstein ambiguity set $\mathcal{P}$ in (3) and encoding each normal distribution $\mathbb{Q} \in \mathcal{P}$ by its mean vector $c \in \mathbb{R}^d$ and covariance matrix $S \in \mathbb{S}_+^d$, we can use Proposition 2.2 to reformulate (A.1b) as

$$\sup \quad \inf_{G,g} \left\langle I_n, S_{xx} + c_x c_x^\top \right\rangle + \left\langle G^\top G, S_{yy} + c_y c_y^\top \right\rangle - \left\langle G, S_{xy} + c_x c_y^\top \right\rangle$$
$$- \left\langle G^\top, S_{yx} + c_y c_x^\top \right\rangle + 2 \left\langle g, G c_y - c_x \right\rangle + g^\top g$$

$$\text{s.t.} \quad c \in \mathbb{R}^d, \quad c_x \in \mathbb{R}^n, \quad c_y \in \mathbb{R}^m$$
$$S \in \mathbb{S}_+^d, \quad S_{xx} \in \mathbb{S}_+^n, \quad S_{yy} \in \mathbb{S}_+^m, \quad S_{xy} = S_{yx}^\top \in \mathbb{R}^{n \times m} \tag{A.2a}$$
$$c = \begin{bmatrix} c_x \\ c_y \end{bmatrix}, \ S = \begin{bmatrix} S_{xx} & S_{xy} \\ S_{yx} & S_{yy} \end{bmatrix} \succeq 0$$
$$\|c - \mu\|^2 + \operatorname{Tr}\left[ S + \Sigma - 2\left( \Sigma^{\frac{1}{2}} S \Sigma^{\frac{1}{2}} \right)^{\frac{1}{2}} \right] \leq \rho^2.$$

Solving the inner minimization problem over $g$ analytically and substituting the optimal solution $g^\star = c_x - G c_y$ back into the objective function shows that (A.2a) is equivalent to

$$\sup \quad \inf_G \left\langle \begin{bmatrix} I_n & -G \\ -G^\top & G^\top G \end{bmatrix}, S \right\rangle$$
$$\text{s.t.} \quad c \in \mathbb{R}^d, \quad S \in \mathbb{S}_+^d \tag{A.2b}$$
$$\|c - \mu\|^2 + \operatorname{Tr}\left[ S + \Sigma - 2\left( \Sigma^{\frac{1}{2}} S \Sigma^{\frac{1}{2}} \right)^{\frac{1}{2}} \right] \leq \rho^2.$$

The minimization over $G$ may now be interchanged with the maximization over $c$ and $S$ by using the classical minimax theorem [2, Proposition 5.5.4], which applies because $c$ and $S$ range over a compact feasible set. The inner maximization problem over $c$ is then solved by $c^\star = \mu$, which maximizes the slack of the Wasserstein constraint. Thus, the minimax problem (A.2b) simplifies to

$$\inf_G \quad \sup_{S \succeq 0} \quad \left\langle \begin{bmatrix} I_n & -G \\ -G^\top & G^\top G \end{bmatrix}, S \right\rangle$$
$$\text{s.t.} \quad \operatorname{Tr}\left[ S + \Sigma - 2\left( \Sigma^{\frac{1}{2}} S \Sigma^{\frac{1}{2}} \right)^{\frac{1}{2}} \right] \leq \rho^2. \tag{A.2c}$$

Assigning a Lagrange multiplier $\gamma \geq 0$ to the Wasserstein constraint and dualizing the inner maximization problem yields

$$\inf_{G} \inf_{\gamma \geq 0} \sup_{S \succeq 0} \left\langle \begin{bmatrix} I_n & -G \\ -G^\top & G^\top G \end{bmatrix}, S \right\rangle + \gamma \left( \rho^2 - \mathrm{Tr}\left[ S + \Sigma - 2\left( \Sigma^{\frac{1}{2}} S \Sigma^{\frac{1}{2}} \right)^{\frac{1}{2}} \right] \right). \tag{A.2d}$$

Strong duality holds because $S = \Sigma \succ 0$ represents a Slater point for the primal maximization problem. Finally, by using Lemma A.1, problem (A.2d) can be reformulated as

$$\begin{aligned}
\inf \quad & \gamma\left(\rho^2 - \mathrm{Tr}\left[\Sigma\right]\right) + \gamma^2 \left\langle (\gamma I_d - [I_n, \ -G]^\top [I_n, \ -G])^{-1}, \Sigma \right\rangle \\
\text{s.t.} \quad & G \in \mathbb{R}^{n \times m}, \quad \gamma \in \mathbb{R}_+ \\
& \gamma I_d \succ [I_n, \ -G]^\top [I_n, \ -G].
\end{aligned} \tag{A.3}$$

By construction, the optimal value of (A.3) provides a lower bound on that of the minimax problem (2). Next, we construct an upper bound by restricting $\mathcal{L}$ to the class of affine estimators.

$$\inf_{\psi \in \mathcal{L}} \sup_{\mathbb{Q} \in \mathcal{P}} \mathbb{E}^{\mathbb{Q}}\left[\|x - \psi(y)\|^2\right] \leq \inf_{G,g} \sup_{\mathbb{Q} \in \mathcal{P}} \mathbb{E}^{\mathbb{Q}}\left[\|x - Gy - g\|^2\right] \tag{A.4}$$

As $\mathcal{P}$ is non-convex, we cannot simply use Sion's minimax theorem to show that the right-hand side of (A.4) equals (A.1b). Instead, we need a more involved argument. Recalling the definition of $\mathcal{P}$ in (3) and encoding each normal distribution $\mathbb{Q} \in \mathcal{P}$ by its mean vector $c \in \mathbb{R}^d$ and covariance matrix $S \in \mathbb{S}_+^d$, we can use Proposition 2.2 to reformulate the right-hand side of (A.4) as

$$\begin{aligned}
\inf_{G,g} \quad \sup \quad & \left\langle I_n, S_{xx} + c_x c_x^\top \right\rangle + \left\langle G^\top G, S_{yy} + c_y c_y^\top \right\rangle - \left\langle G, S_{xy} + c_x c_y^\top \right\rangle \\
& \quad - \left\langle G^\top, S_{yx} + c_y c_x^\top \right\rangle + 2\left\langle g, G c_y - c_x \right\rangle + g^\top g \\[4pt]
\text{s.t.} \quad & c \in \mathbb{R}^d, \quad c_x \in \mathbb{R}^n, \quad c_y \in \mathbb{R}^m \\
& S \in \mathbb{S}_+^d, \quad S_{xx} \in \mathbb{S}_+^n, \quad S_{yy} \in \mathbb{S}_+^m, \quad S_{xy} = S_{yx}^\top \in \mathbb{R}^{n \times m} \\
& c = \begin{bmatrix} c_x \\ c_y \end{bmatrix}, \ S = \begin{bmatrix} S_{xx} & S_{xy} \\ S_{yx} & S_{yy} \end{bmatrix} \succeq 0 \\
& \|c - \mu\|^2 + \mathrm{Tr}\left[ S + \Sigma - 2\left( \Sigma^{\frac{1}{2}} S \Sigma^{\frac{1}{2}} \right)^{\frac{1}{2}} \right] \leq \rho^2.
\end{aligned} \tag{A.5a}$$

Next, we introduce the set $\mathcal{C} \triangleq \{c \in \mathbb{R}^d : \|c - \mu\| \leq \rho\}$ as well as the auxiliary functions

$$D(G) \triangleq \begin{bmatrix} I_n & -G \\ -G^\top & G^\top G \end{bmatrix} \quad \text{and} \quad b(G,g) \triangleq \begin{bmatrix} -g \\ G^\top g \end{bmatrix}$$

to reformulate problem (A.5a) as

$$\begin{aligned}
\inf_{G,g} \quad \sup_{\substack{c \in \mathcal{C} \\ S \succeq 0}} \quad & \left\langle D(G), S + c\,c^\top \right\rangle + 2\left\langle b(G,g), c \right\rangle + g^\top g \\
\text{s.t.} \quad & \|c - \mu\|^2 + \mathrm{Tr}\left[ S + \Sigma - 2\left( \Sigma^{\frac{1}{2}} S \Sigma^{\frac{1}{2}} \right)^{\frac{1}{2}} \right] \leq \rho^2.
\end{aligned} \tag{A.5b}$$

We emphasize that the constraint $c \in \mathcal{C}$ is redundant in (A.5b) but will facilitate further simplifications below. Note also that $D(G) \succeq 0$, and thus the minimax problem (A.5b) involves a cumbersome convex maximization problem over $c$. By employing a penalty formulation of the Wasserstein constraint, the inner maximization problem over $c$ and $S$ in (A.5b) can be re-expressed as

$$\begin{aligned}
\sup_{\substack{c \in \mathcal{C} \\ S \succeq 0}} \inf_{\gamma \geq 0} \quad & \left\langle D(G), S + c\,c^\top \right\rangle + 2\left\langle b(G,g), c \right\rangle + g^\top g \\
& + \gamma \left( \rho^2 - \|c - \mu\|^2 + \mathrm{Tr}\left[ S + \Sigma - 2\left( \Sigma^{\frac{1}{2}} S \Sigma^{\frac{1}{2}} \right)^{\frac{1}{2}} \right] \right).
\end{aligned}$$

Here, the minimization over $\gamma$ and the maximization over $S$ may be interchanged by strong duality, which holds because $S = \Sigma \succ 0$ constitutes a Slater point for the primal problem, see, *e.g.*, [2, Proposition 5.3.1]. We note that when $\|c - \mu\| = \rho$, the feasible set of $S$ reduces to a singleton, and

thus strong duality holds trivially. The emerging inner maximization problem over $S$ can then be solved analytically by using Lemma A.1. In summary, the minimax problem (A.5b) is equivalent to

$$\inf_{G,g} \sup_{c \in \mathcal{C}} \inf_{\gamma \geq 0} \quad \langle D(G), cc^\top \rangle + 2\langle b(G,g), c \rangle + g^\top g + \gamma \left( \rho^2 - \|c - \mu\|^2 - \mathrm{Tr}\left[\Sigma\right] \right)$$
$$+ \gamma^2 \langle (\gamma I_d - D(G))^{-1}, \Sigma \rangle \qquad (A.5c)$$
$$\text{s.t.} \quad \gamma I_d \succ D(G).$$

Observe now that the optimal value function of the innermost minimization problem over $\gamma$ in (A.5c) is convex in $g$ and, thanks to the constraint $\gamma I_d - D(G) \succ 0$, concave in $c$ for every fixed $G$. By the classical minimax theorem [2, Proposition 5.5.4], which applies because $c$ ranges over the compact set $\mathcal{C}$, we may thus interchange the infimum over $g$ with the supremum over $c$. After replacing $D(G)$ and $b(G,g)$ with their definitions, it becomes clear that the innermost minimization problem over $g$ admits the analytical solution $g^\star = \mu_x - G\mu_y$. Thus, problem (A.5c) is equivalent to

$$\inf_{G} \sup_{c \in \mathcal{C}} \inf_{\gamma \geq 0} \quad \gamma \left( \rho^2 - \|c - \mu\|^2 - \mathrm{Tr}\left[\Sigma\right] \right) + \gamma^2 \langle (\gamma I_d - [I_n, \ -G]^\top [I_n, \ -G])^{-1}, \Sigma \rangle$$
$$\text{s.t.} \quad \gamma I_d \succ [I_n, \ -G]^\top [I_n, \ -G].$$
$$(A.5d)$$

By invoking the minimax theorem [2, Proposition 5.5.4] once again, the inner infimum over $\gamma$ can be interchanged with the supremum over $c$. As the resulting inner maximization problem over $c$ is solved by $c^\star = \mu$, problem (A.5d) is thus equivalent to (A.3). In summary, we have shown that (A.3) provides both an upper bound on the left-hand side of (A.1) as well as a lower bound on the right-hand side of (A.1). Thus, the inequality in (A.1) is in fact an equality. $\qquad \square$

## A.2   Proof of Theorem 2.5

The proof of Theorem 2.5 relies on the following lemma, which extends a similar result from [6].

**Lemma A.2** (Analytical solution of direction-finding subproblem)**.** For any fixed $\Sigma \in \mathbb{S}_{++}^d$ and $D \in \mathbb{S}_+^d \backslash \{0\}$, the optimization problem

$$\sup_{S \in \mathbb{S}_+^d} \quad \langle S, D \rangle$$
$$\text{s.t.} \quad \mathrm{Tr}\left[ S + \Sigma - 2 \left( \Sigma^{\frac{1}{2}} S \Sigma^{\frac{1}{2}} \right)^{\frac{1}{2}} \right] \leq \rho^2$$

is solved by

$$S^\star = (\gamma^\star)^2 \, (\gamma^\star I_d - D)^{-1} \Sigma (\gamma^\star I_d - D)^{-1},$$

where $\gamma^\star$ is the unique solution with $\gamma^\star I_d \succ D$ of the algebraic equation

$$\rho^2 - \langle \Sigma, \left( I_d - \gamma^\star (\gamma^\star I_d - D)^{-1} \right)^2 \rangle = 0.$$

Moreover, we have $S^\star \succeq \underline{\sigma} I_d$, where $\underline{\sigma} \triangleq \lambda_{\min}(\Sigma)$.

*Proof of Lemma A.2.* The optimality of $S^\star$ follows immediately from [6, Theorem 5.1]. Moreover, the spectral norm of $(S^\star)^{-1}$ obeys the following estimate.

$$\|(S^\star)^{-1}\| \leq \|I_d - \frac{1}{\gamma^\star} D\| \cdot \|\Sigma^{-1}\| \cdot \|I_d - \frac{1}{\gamma^\star} D\| \leq \|\Sigma^{-1}\| = \underline{\sigma}^{-1}$$

As the largest eigenvalue of $(S^\star)^{-1}$ is bounded by $\underline{\sigma}^{-1}$, we may conclude that $S^\star \succeq \underline{\sigma} I_d$. $\qquad \square$

*Proof of Theorem 2.5.* The proof of Theorem 2.3 has shown that the original infinite-dimensional minimax problem (2) is equivalent to the finite-dimensional minimax problem (A.2c). By Lemma A.2, the solution of the inner maximization problem in (A.2c) satisfies $S^\star \succeq \underline{\sigma} I_d$. Thus, one may append the redundant constraint $S \succeq \underline{\sigma} I_d$ to this inner problem without sacrificing optimality. By interchanging the minimization over $G$ with the maximization over $S$, which is allowed by [2, Proposition 5.5.4], problem (A.2c) can thus be reformulated as

$$\sup_{S \succeq 0} \inf_{G} \quad \langle \begin{bmatrix} I_n & -G \\ -G^\top & G^\top G \end{bmatrix}, S \rangle$$
$$\text{s.t.} \quad \mathrm{Tr}\left[ S + \Sigma - 2 \left( \Sigma^{1/2} S \Sigma^{1/2} \right)^{1/2} \right] \leq \rho^2 \qquad (A.6)$$
$$S \succeq \underline{\sigma} I_d.$$

Recall that $\sigma > 0$, which implies that $S \succ 0$. Hence, the unconstrained quadratic minimization problem over $G$ in (A.6) has a unique solution $G^\star$, which can be obtained analytically by solving the problem's first-order optimality condition. Specifically, we have

$$2G^\star S_{yy} - 2S_{xy} = 0 \quad \Longleftrightarrow \quad G^\star = S_{xy}S_{yy}^{-1}.$$

Substituting $G^\star$ into (A.6) yields the desired maximization problem (5). By construction, this convex program is equivalent to nature's decision problem on the right-hand side of (4), and thus it is easy to see that the least favorable prior is given by $\mathbb{Q}^\star = \mathcal{N}_d(\mu, S^\star)$. Next, we solve the Bayesian estimation problem

$$\inf_{\psi \in \mathcal{L}} \mathbb{E}^{\mathbb{Q}^\star}\left[\|x - \psi(y)\|^2\right].$$

An elementary analytical calculation reveals that this problem is solved by $\psi^\star(y) = S_{xy}^\star(S_{yy}^\star)^{-1}(y - \mu_y) + \mu_x$. Moreover, this solution is unique because $S^\star \succeq \underline{\sigma}I_d$, which implies that the objective function is strictly convex. By Theorem 2.3 and [3, Section 5.5.5], we may then conclude that $\psi^\star$ is also optimal in (2). This observation completes the proof. $\qquad\square$

## A.3 Proof of Theorem 3.2

The following lemma suggests upper and lower bounds on the (unique) root $\gamma^\star$ of the function $h(\gamma)$ defined in (8). Note that this root is computed approximately using bisection in Algorithm 1.

**Lemma A.3** (Bisection interval). *For any $\rho > 0$, the solution of the algebraic equation $h(\gamma^\star) = 0$ resides in the interval $[\gamma_{\min}, \gamma_{\max}]$, where*

$$\gamma_{\min} \triangleq \lambda_1\left(1 + \sqrt{v_1^\top \Sigma v_1 / \rho}\right), \quad \gamma_{\max} \triangleq \lambda_1\left(1 + \sqrt{\mathrm{Tr}\,[\Sigma]/\rho}\right), \qquad (A.7)$$

*the scalar $\lambda_1$ is the largest eigenvalue of $D \triangleq \nabla f(S)$, and $v_1$ is a corresponding eigenvector.*

*Proof of Lemma A.3.* Let $D = \sum_{i=1}^d \lambda_i v_i v_i^\top$ be the spectral decomposition of $D$. The function $h$ can be equivalently rewritten as

$$\rho^2 - \sum_{i=1}^d \left(\frac{\lambda_i}{\gamma - \lambda_i}\right)^2 v_i^\top \Sigma v_i,$$

where the summation admits the following bounds:

$$\left(\frac{\lambda_1}{\gamma - \lambda_1}\right)^2 v_1^\top \Sigma v_1 \leq \sum_{i=1}^d \left(\frac{\lambda_i}{\gamma - \lambda_i}\right)^2 v_i^\top \Sigma v_i \leq \left(\frac{\lambda_1}{\gamma - \lambda_1}\right)^2 \mathrm{Tr}\,[\Sigma].$$

Equating the two bounds to $\rho^2$ yields $\gamma_{\min}$ and $\gamma_{\max}$, respectively. $\qquad\square$

*Proof of Theorem 3.2.* The proof of Lemma A.2 implies that $L(\gamma) \triangleq \gamma^2(\gamma I_d - D)^{-1}\Sigma(\gamma I_d - D)^{-1}$ is feasible in (7b) for every $\gamma$ with $\gamma I_d \succ D$ and $h(\gamma) > 0$; see [4, Theorem 2]. Moreover, $L(\gamma^\star)$ is optimal in (7b) if $\gamma^\star I_d \succ D$ and $h(\gamma^\star) = 0$. Algorithm 1 uses a bisection procedure to compute an approximation $\gamma$ of $\gamma^\star$ such that $L(\gamma)$ is feasible and $\varepsilon$-suboptimal in (7b). The degree of suboptimality of $L(\gamma)$ equals $\langle L(\gamma^\star) - L(\gamma), D\rangle$. The true optimal value $\langle L(\gamma^\star), D\rangle$ is inaccessible but can be estimated above by the objective value of $\gamma$ in the Lagrangian dual of (7b), which can be expressed as

$$\min_{\gamma:\, \gamma I_d \succ D} \gamma(\rho^2 - \mathrm{Tr}\,[\Sigma]) + \gamma^2\langle(\gamma I_d - D)^{-1}, \Sigma\rangle,$$

see also [6, Proposition 2.8]. Thus, the suboptimality of $L(\gamma)$ is bounded above by

$$\langle L(\gamma^\star) - L(\gamma), D\rangle \leq \gamma(\rho^2 - \mathrm{Tr}\,[\Sigma]) + \gamma^2\langle(\gamma I_d - D)^{-1}, \Sigma\rangle - \langle L(\gamma), D\rangle.$$

Lemma A.3 ensures that $\gamma^\star \in [\gamma_{\min}, \gamma_{\max}]$, and therefore it suffices to search over this interval. Note that the function $h$ is Lipschitz continuous in the bisection interval, and therefore, the bisection Algorithm 1 will terminate in finite time. $\qquad\square$

## A.4 Proof of Theorem 3.3

The proof of Theorem 3.3 widely parallels that of [5, Theorem 1]. The key ingredient is to prove that the *curvature constant* of the problem's (negative) objective function $-f$ is bounded.

**Definition A.4** (Curvature constant). The curvature constant $C_g$ of the convex function $g$ with respect to a compact domain $\mathcal{S}$ is defined as

$$
C_g \triangleq \begin{cases}
\sup\limits_{X,Y,Z,\alpha} & \frac{2}{\alpha^2} \left( g(Z) - g(X) - \langle Z - X, \nabla g(X) \rangle \right) \\
\text{s.t.} & Z = (1-\alpha)X + \alpha Y \\
& X, Y \in \mathcal{S}, \quad \alpha \in [0,1].
\end{cases}
$$

In order to bound the curvature constant of $-f$, we need several preparatory lemmas.

**Lemma A.5** ([1, Fact 7.4.9]). For any $A \in \mathbb{R}^{n \times m}, B \in \mathbb{R}^{m \times l}, C \in \mathbb{R}^{l \times k}$, and $D \in \mathbb{R}^{k \times n}$, we have

$$
\text{Tr}\,[ABCD] = \text{vec}(A)^\top (B \otimes D^\top)\,\text{vec}(C^\top),
$$

where '$\otimes$' stands for the Kronecker product, while '$\text{vec}(\cdot)$' denotes the vectorization of a matrix.

**Lemma A.6** (Bounded feasible set). If $S$ is feasible in (5), then $S \preceq \bar{\sigma} I_d$, where $\bar{\sigma} \triangleq (\rho + \sqrt{\text{Tr}\,[\Sigma]})^2$.

*Proof of Lemma A.6.* We seek an upper bound on the maximum eigenvalue of $S$ uniformly across all covariance matrices $S$ feasible in (5), that is, we seek an upper bound on the optimal value of

$$
\begin{aligned}
\sup_{S \succeq 0} \quad & \|S\| \\
\text{s.t.} \quad & \text{Tr}\left[ S + \Sigma - 2 \left( \Sigma^{\frac{1}{2}} S \Sigma^{\frac{1}{2}} \right)^{\frac{1}{2}} \right] \leq \rho^2.
\end{aligned} \tag{A.8}
$$

Problem (A.8) is a non-convex optimization problem because we maximize a convex function (the spectral norm of $S$) over a convex set. An easily computable upper bound is obtained by solving

$$
\begin{aligned}
\sup_{S \succeq 0} \quad & \langle S, I_d \rangle \\
\text{s.t.} \quad & \text{Tr}\left[ S + \Sigma - 2 \left( \Sigma^{\frac{1}{2}} S \Sigma^{\frac{1}{2}} \right)^{\frac{1}{2}} \right] \leq \rho^2.
\end{aligned} \tag{A.9}
$$

Indeed, note that $\text{Tr}\,[S] = \langle S, I_d \rangle \geq \|S\|$, where the inequality holds because $S \succeq 0$. By Lemma A.2, which studies a more general problem with an arbitrary linear objective function $\langle S, D \rangle$, problem (A.9) has an analytical solution that is found by solving the following algebraic equation in $\gamma$.

$$
\rho^2 - \left\langle \Sigma, \left( I_d - \gamma^\star(\gamma^\star I_d - I_d)^{-1} \right)^2 \right\rangle = 0 \iff \rho^2 - \left( \frac{1}{\gamma^\star - 1} \right)^2 \text{Tr}\,[\Sigma] = 0
$$

In the special case considered here, this equation can be solved in closed form, and there is no need for a bisection algorithm. Specifically, we have $\gamma^\star = 1 + \sqrt{\text{Tr}\,[\Sigma]}/\rho$, and thus (A.9) is solved by

$$
S^\star = (\gamma^\star)^2 (\gamma^\star I_d - I_d)^{-1} \Sigma (\gamma^\star I_d - I_d)^{-1} = \left( \frac{\gamma^\star}{\gamma^\star - 1} \right)^2 \Sigma = \frac{(\rho + \sqrt{\text{Tr}\,[\Sigma]})^2}{\text{Tr}\,[\Sigma]} \Sigma.
$$

Therefore, problem (A.8) is upper bounded by $\text{Tr}\,[S^\star] = (\rho + \sqrt{\text{Tr}\,[\Sigma]})^2$. $\qquad\square$

For ease of exposition, we now define the (compact) feasible set of problem (5) as

$$
\mathcal{S} \triangleq \left\{ S \in \mathbb{S}_+^d : \text{Tr}[S + \Sigma - 2(\Sigma^{\frac{1}{2}} S \Sigma^{\frac{1}{2}})^{\frac{1}{2}}] \leq \rho^2, \quad S \succeq \underline{\sigma} I_d \right\} \tag{A.10}
$$

**Lemma A.7** (Curvature bound). The curvature constant $C_{-f}$ of the (negative) objective function $-f$ over the feasible set $\mathcal{S}$ satisfies $C_{-f} \leq \overline{C} \triangleq 2\bar{\sigma}^4 / \underline{\sigma}^3$.

*Proof of Lemma A.7.* We first expand the negative objective function $-f$ at $S \in \mathbb{S}_+^d$. By Lemma A.5, for any symmetric perturbation matrix $\Delta$ with a characteristic block structure of the form

$$
\Delta = \begin{bmatrix} \Delta_{xx} & \Delta_{xy} \\ \Delta_{xy}^\top & \Delta_{yy} \end{bmatrix} \in \mathbb{S}^d,
$$

the negative objective function $-f(S + \Delta)$ can be expressed as

$$\mathrm{Tr}\left[-S_{xx} - \Delta_{xx} + (S_{xy} + \Delta_{xy})(S_{yy} + \Delta_{yy})^{-1}(S_{yx} + \Delta_{xy}^{\top})\right]$$
$$= \mathrm{Tr}\left[-S_{xx} - \Delta_{xx}\right] +$$
$$\qquad \mathrm{Tr}\left[(S_{xy} + \Delta_{xy})S_{yy}^{-1}(I_m - \Delta_{yy}S_{yy}^{-1} + (\Delta_{yy}S_{yy}^{-1})^2 + \mathcal{O}(\|\Delta_{yy}\|^3))(S_{yx} + \Delta_{xy}^{\top})\right]$$
$$= \mathrm{Tr}\left[-S_{xx} + S_{xy}S_{yy}^{-1}S_{yx}\right] - \mathrm{Tr}\left[\Delta_{xx} - \Delta_{xy}S_{yy}^{-1}S_{yx} + S_{xy}S_{yy}^{-1}\Delta_{yy}S_{yy}^{-1}S_{yx} - S_{xy}S_{yy}^{-1}\Delta_{xy}^{\top}\right]$$
$$\quad - \mathrm{Tr}\left[\Delta_{xy}S_{yy}^{-1}\Delta_{yy}S_{yy}^{-1}S_{yx} - \Delta_{xy}S_{yy}^{-1}\Delta_{xy}^{\top} + S_{xy}S_{yy}^{-1}\Delta_{yy}S_{yy}^{-1}\Delta_{xy}^{\top} - S_{xy}S_{yy}^{-1}(\Delta_{yy}S_{yy}^{-1})^2 S_{yx}\right]$$
$$\quad + \mathcal{O}(\|\Delta\|^3)$$

$$= \mathrm{Tr}\left[-S_{xx} + S_{xy}S_{yy}^{-1}S_{yx}\right] - \langle D, \Delta \rangle + \frac{1}{2}\begin{bmatrix}\mathrm{vec}\,\Delta_{xx} \\ \mathrm{vec}\,\Delta_{xy} \\ \mathrm{vec}\,\Delta_{yy}\end{bmatrix}^{\top} H \begin{bmatrix}\mathrm{vec}\,\Delta_{xx} \\ \mathrm{vec}\,\Delta_{xy} \\ \mathrm{vec}\,\Delta_{yy}\end{bmatrix} + \mathcal{O}(\|\Delta\|^3),$$

where

$$D = \begin{bmatrix} I_n & -S_{xy}S_{yy}^{-1} \\ -S_{yy}^{-1}S_{yx} & S_{yy}^{-1}S_{yx}S_{xy}S_{yy}^{-1} \end{bmatrix} \in \mathbb{S}_+^d \tag{A.11}$$

and

$$H = \begin{bmatrix} 0 & 0 & 0 \\ 0 & 2S_{yy}^{-1} \otimes I_n & -2S_{yy}^{-1} \otimes S_{xy}S_{yy}^{-1} \\ 0 & -2S_{yy}^{-1} \otimes S_{yy}^{-1}S_{yx} & 2S_{yy}^{-1} \otimes S_{yy}^{-1}S_{yx}S_{xy}S_{yy}^{-1} \end{bmatrix} \in \mathbb{S}_+^{(n^2+nm+m^2)}.$$

Note that the matrix $D$ represents the gradient of $f$, which plays a crucial role in the Frank-Wolfe algorithm. Similarly, $H$ can be viewed as a compressed version of the Hessian matrix of $-f$, where the redundant rows and columns corresponding to $S_{yx}$ have been eliminated. Thus, the Lipschitz constant of the gradient $\nabla f$ can be upper bounded by the largest eigenvalue of $H$, which is given by

$$\|H\| = 2\|S_{yy}^{-1} \otimes D\| = 2\|S_{yy}^{-1}\| \cdot \|D\|. \tag{A.12}$$

By a standard Schur complement argument, we then have

$$S = \begin{bmatrix} S_{xx} & S_{xy} \\ S_{yx} & S_{yy} \end{bmatrix} = \begin{bmatrix} I_n & S_{xy}S_{yy}^{-1} \\ 0 & I_m \end{bmatrix} \begin{bmatrix} S_{xx} - S_{xy}S_{yy}^{-1}S_{yx} & 0 \\ 0 & S_{yy} \end{bmatrix} \begin{bmatrix} I_n & 0 \\ S_{yy}^{-1}S_{yx} & I_m \end{bmatrix}.$$

Next, define the set

$$\mathcal{V} \triangleq \left\{ z = [x^{\top}, y^{\top}]^{\top} \in \mathbb{R}^d : S_{yy}^{-1}S_{yx}\,x + y = 0 \right\},$$

and note that any $z \in \mathcal{V}$ satisfies

$$\begin{bmatrix} I_n & 0 \\ S_{yy}^{-1}S_{yx} & I_m \end{bmatrix} z = \begin{bmatrix} x \\ 0 \end{bmatrix}.$$

Thus, by the definition of the smallest eigenvalue, we have

$$\lambda_{\min}(S) = \min_{z \neq 0} \frac{z^{\top}Sz}{z^{\top}z} \leq \min_{\substack{z \neq 0 \\ z \in \mathcal{V}}} \frac{z^{\top}Sz}{z^{\top}z} \leq \lambda_{\min}(S_{xx} - S_{xy}S_{yy}^{-1}S_{yx}) \implies S_{xx} - S_{xy}S_{yy}^{-1}S_{yx} \succeq \underline{\sigma}I_n.$$

Moreover, by the Cauchy interlacing theorem [1, Theorem 8.4.5], Lemma A.6, and basic properties of the spectral norm, we have

$$\|S_{yy}^{-1}\| \leq \|S^{-1}\| \leq \frac{1}{\underline{\sigma}}, \quad \|S_{xx}\| \leq \bar{\sigma} \quad \text{and} \quad \|S_{yy}\| \leq \bar{\sigma}.$$

Using the above inequalities, one can show that

$$\frac{1}{\bar{\sigma}}I_m \preceq S_{yy}^{-1} \implies S_{xy}S_{yx} \preceq \bar{\sigma}S_{xy}S_{yy}^{-1}S_{yx}$$

and

$$\bar{\sigma}I_n \succeq S_{xx} \succeq S_{xx} - S_{xy}S_{yy}^{-1}S_{yx} \succeq \underline{\sigma}I_n \implies S_{xy}S_{yy}^{-1}S_{yx} \preceq (\bar{\sigma} - \underline{\sigma})\,I_n.$$

Setting $B = [I_n, -S_{xy}S_{yy}^{-1}]$, the above inequalities imply that

$$\|D\| = \|B^\top B\| = \|BB^\top\| = \|I_n + S_{xy}S_{yy}^{-2}S_{yx}\| = 1 + \|S_{xy}S_{yy}^{-2}S_{yx}\|$$
$$= 1 + \|S_{yy}^{-1}S_{yx}S_{xy}S_{yy}^{-1}\|$$
$$\leq 1 + \|S_{yy}^{-1}\|^2 \cdot \|S_{yx}S_{xy}\|$$
$$\leq 1 + \frac{\bar{\sigma}(\bar{\sigma} - \underline{\sigma})}{\underline{\sigma}^2} \leq \frac{\bar{\sigma}^2}{\underline{\sigma}^2}.$$

By combining the last estimate with (A.12), we then find that the Lipschitz constant of $\nabla f$ satisfies

$$\text{Lip}(\nabla f) = \|H\| \leq \frac{2\bar{\sigma}^2}{\underline{\sigma}^3}.$$

The diameter of the feasible set $\mathcal{S}$ with respect to the Frobenius norm satisfies

$$\text{diam}(\mathcal{S}) = \sup_{S_1, S_2 \in \mathcal{S}} \|S_1 - S_2\|_F \leq \sup_{S_1, S_2 \in \mathcal{S}} \text{Tr}\,[S_1 - S_2] \leq \sup_{S \in \mathcal{S}} \text{Tr}\,[S] \leq \bar{\sigma},$$

where the first inequality holds due to [1, Equation (9.2.16)], and the last inequality follows from the proof of Lemma A.6. Therefore, by [5, Lemma 7], the curvature constant $C_{-f}$ admits the estimate

$$C_{-f} \leq (\text{diam}(\mathcal{S}))^2 \text{Lip}(\nabla f) \leq \frac{2\bar{\sigma}^4}{\underline{\sigma}^3}.$$

This observation completes the proof. □

*Proof of Theorem 3.3.* By Lemma A.7, the curvature constant $C_{-f}$ is bounded, and thus one can directly apply [5, Theorem 1] to find the convergence rate. □

## Appendix B Sequential versus Static Estimation

We have resolved the filtering problem underlying Figure 4(c) as a single (static) estimation problem in the spirit of Section 2, where the entire observation history $Y_t \triangleq (y_1, \ldots, y_1)$ is interpreted as a single observation used to predict $x_t$. To our surprise, we found that the sequential filtering approach advocated in Section 4 outperforms this alternative static approach even if an oracle reveals the optimal radius of the ambiguity set (for $t = 100$, *e.g.*, the static estimation error is 37.5 dB, while the sequential estimation error is only 24.5 dB). In fact, for the static estimation problem the optimal radius of the Wasserstein ball is $\rho = 0$ whenever $t \geq 5$, that is, robustification does not improve performance. In contrast, in the sequential filtering approach robustification always helps. A possible explanation for this observation is that in the static approach our lack of information about the system uncertainty propagates through the dynamics. As such, it renders robust estimation ineffective when applied globally to the entire observation history at once. In contrast, in the sequential approach the robustification at each stage appears to limit such an uncertainty propagation.