[Reviews · NeurIPS 2018]

Reviewer 1



This paper proposes to use Wasserstein ambiguity sets to model uncertainty in the state estimation problem. The most relevant literature is H infinity filtering, where we solve "minimax" filtering optimization subject to the various disturbance (noise) models. While the typical H infinity filtering approach seeks a worst case disturbance "point", this paper seeks a worst case disturbance "distributions", which is more general then the usual H infinity filtering setting. I think the idea is new and interesting. Furthermore the paper proposes a new efficient algorithm for designing a filter under this setup, which is also interesting by its own.

Reviewer 2



SUMMARY: The authors address the robust regression problem over a Wasserstein ambiguity set, in line with several recent papers such as the cited papers [12, 15, 16] and also the recent paper “Certifying Some Distributional Robustness with Principled Adversarial Training“ (Sinha et al. 2018). In this study, the authors focus on a restricted but important scenario: they consider the least-square regression problem and the ambiguity sets is made Gaussian measures in the Wasserstein ball of a nominal Gaussian measure. In particular, this important case allows them provides to devise a distributiuonally-robust Kalman Filter. They show that the problem is convex and can be solved efficiently with a Frank-Wolf algorithm. The advantage of the method over non-robust Bayes estimation is demonstrated on synthetic data. Quality: The paper is technically sound. Although the problem addresses an apparently simple case of Gaussian Wasserstein ambiguity set, the resulting optimization is non-trivial (non-linear convex SDP). The authors manage to derive an efficient Franck-Wolf algorithm together with convergences guarantees in in Theorem 3.3. The distributionally-robust Kalman filter is well-described and is a natural application of the proposed problem. It would have been very nice to see some experiments on real data rather than only synthetic data. Clarity: The paper is overall well-written, although the introduction is not really clear for a reader who is not expert in the field of robust regression. I believe the paper would gain a lot by adding some simple visualizations illustrating the problem. For instance, it would be nice for instance to have some visualizations of nominal distribution P together with optimal \Psi and Q in some simple scenarios such as x and y being one-dimensional data. This may also let the reader understand intuitively better the difference in behavior between Wasserstein and tau-divergence ambiguity sets. Originality and Significance: The paper addresses a particular case of the robust regression over Wasserstein ambiguity sets, when the ambiguity sets is restricted over Gaussian measures. As far as I know, this vanilla case has not been treated in the literature so far but I believe it is an important case to understand and tp be able to solve efficiently. The application to the distributionally-robust Kalman filter illustrates in particular the importance of this problem.

Reviewer 3



Update: I have read the authors' response with interest; they have addressed most of my comments well. Regarding comment (2), I do not not see why one would need a formula for the Wasserstein distance between normal and arbitrary non-normal distributions. For any distribution Q, a normal distribution that has the same first and second moments as Q must be closer than Q to the nominal distribution in Wasserstein distance, as there exists a joint Gaussian distribution with the same second moment as the joint distribution attaining the Wasserstein distance between Q and the nominal distribution. Since a normal distribution with the same first and second moments will have linear estimation error identical to that of Q, this shows we may restrict our attention to normal distributions in the uncertainty set. --- The authors study robust estimation with a mean squared-error objective and uncertainty set defined by the Wasserstein distance with respect to a quadratic cost function. They show how to find the robust estimator by solving a convex program, give a Frank-Wolfe method for solving the program, and prove convergence guarantees for the method. Based on these developments, the authors propose a robust variant of the Kalman filter. In my opinion this is a very good paper that contributes substantially to our knowledge on distributionally robust optimization with Wasserstein uncertainty, in which there seems to be fairly widespread interest among the NIPS community. As such, I am strongly in favor of accepting this paper. Below I list some comments and suggestions regarding the paper. 1) While your proposal of a robust Kalman filter is quite appealing, I do not think it merits the title "minimax", that you attribute to it in the abstract and throughout the paper. To the best of my understanding the derivation of your filter is heuristic and meant to combine your robust MSE estimation formulation with the efficient recursive structure of the classical Kalman filter. As such, there is no uncertainty set for which the proposed filter minimizes the error of the worst-case instance, and therefore it is not "minimax". Please clarify this point. 2) In Section 2 you consider minimax estimators in the set of all measurable functions of y, and a nonconvex uncertainty set consisting of a Wasserstein ball restricted to Gaussian distributions. This seems to give exactly the same result as restricting the set of estimators to affine functions of y, and letting the uncertainty set be simply the Wasserstein ball (considering all distributions). While equivalent, I believe the latter framing is better, as it makes Theorem 2.3 immediate from standard convex-concave optimization results, and it makes the proof of Theorem 2.5 substantially simpler. At the very least, the paper should comment on this alternative formulation. 3) It would be good if you could provide some additional illustration and discussion of what the worst case covariance S^* looks like. A natural test case is Wiener filtering (y = x + n, with n, x independent) - does the worst-case covariance maintain the independent noise structure? Does it have stronger noise, different input covariance, or both? It will be interesting to compare the result to robust Wiener filters with other uncertainty measures (see comment 5 below). 4) A straightforward approach to robust Kalman filtering is to treat estimation of x_t from Y_t simply as a linear estimation problem, and solve it with exactly the same robust formulation you develop in Sections 2 and 3. While far less computationally efficient than the robust filter you propose, this approach has the advantage of actually being minimax in some clear sense. As you seem to be able to run Algorithm 2 with d in the hundreds, it will be interesting to evaluate the simple approach described for the low dimensional test case considered in Section 5.2, and compare it to the proposed method. 5) The paper is missing some references on robust MSE estimation, which has been studied for many years; see "A competitive minimax approach to robust estimation of random parameters" by Eldar and Merhav (2004) and the references therein. These works consider various uncertainty sets for the input covariance, and can therefore be viewed as distributionally robust formulations. The authors might also consider citing more of the recent literature on Wasserstein-distributionally robust optimization, such as Sinha et al. (2018) and Blanchet et al. (2016). Addressing the issues above above will, in my opinion, turn your paper from very good to excellent. Below are a few additional minor comments: - In line 108, replace "efficient" with a more conservative adjective, such as "tractable". - \bar{sigma} is only defined after Algorithm 2 is stated, which is confusing; it should instead be an input to the algorithm. - Using MMSE for "minimax mean squared error" (Section 5.1) is confusing, since MMSE is a commonly-used acronym for "minimum mean squared error". Consider using MMMSE or M3SE instead. - In lines 222-225, \Delta and \Delta^\star appear to be used interchangeably. - In line 97 of the Supplementary Material, I believe it should be Theorem 3.3 rather than Theorem 2.5.